evolution, behaviour

cultural evolution, cultural complexity, multilevel societies, small-world networks, social structure

**Authors for correspondence:**
Mauricio Cantor
e-mail: mcantor@ab.mpg.de
Lucy M. Aplin
e-mail: laplin@ab.mpg.de

†Joint first authors, arranged alphabetically.
‡Joint senior authors.

# Social network architecture and the tempo of cumulative cultural evolution

Mauricio Cantor[1,3,†], Michael Chimento[2,4,5,†], Simeon Q. Smeele[2,6,†], Peng He[4,5,7,8], Danai Papageorgiou[4,5,7,8], Lucy M. Aplin[2,4,‡] and Damien R. Farine[4,5,7,8,‡]

[1]Department for the Ecology of Animal Societies, and [2]Cognitive and Cultural Ecology Research Group, Max Planck Institute of Animal Behavior, Am Obstberg 1, Radolfzell 78315, Konstanz, Germany
[3]Departamento de Ecologia e Zoologia, Universidade Federal de Santa Catarina, Florianópolis, Brazil
[4]Department of Biology, and [5]Centre for the Advanced Study of Collective Behaviour, University of Konstanz, Konstanz, Germany
[6]Department of Human Behavior, Ecology and Culture, Max Planck Institute for Evolutionary Anthropology, Leipzig, Germany
[7]Department of Collective Behaviour, Max Planck Institute of Animal Behavior, Konstanz, Germany
[8]Department of Evolutionary Biology and Environmental Science, University of Zurich, Zurich, Switzerland

MCa, 0000-0002-0019-5106; MCh, 0000-0001-5697-1701; SQS, 0000-0003-1001-6615; DRF, 0000-0003-2208-7613

The ability to build upon previous knowledge—cumulative cultural evolution—is a hallmark of human societies. While cumulative cultural evolution depends on the interaction between social systems, cognition and the environment, there is increasing evidence that cumulative cultural evolution is facilitated by larger and more structured societies. However, such effects may be interlinked with patterns of social wiring, thus the relative importance of social network architecture as an additional factor shaping cumulative cultural evolution remains unclear. By simulating innovation and diffusion of cultural traits in populations with stereotyped social structures, we disentangle the relative contributions of network architecture from those of population size and connectivity. We demonstrate that while more structured networks, such as those found in multilevel societies, can promote the recombination of cultural traits into high-value products, they also hinder spread and make products more likely to go extinct. We find that transmission mechanisms are therefore critical in determining the outcomes of cumulative cultural evolution. Our results highlight the complex interaction between population size, structure and transmission mechanisms, with important implications for future research.

## 1. Background

Cumulative cultural evolution (CCE)—where iterative innovations and social transmission generate cultural accumulation over time [1–3]—is key to humans' ecological success and worldwide distribution [4,5]. While CCE fundamentally depends on the interplay between cognition and social learning mechanisms [1], it is increasingly clear that demography can modulate the rate of cultural evolution [6–9]. Large population sizes [10,11], greater population turnover and more densely connected societies [3,12] can all provide greater innovative potential, more learning models, faster diffusion and reduced extinction risk of useful innovations [7,13–15]. For example, increasing population density as well as the migration of hunter–gatherers during the upper Palaeolithic transition led to the explosion of culture that forms the basis of modern human societies [8,15]. Yet it remains unclear how variation in the wiring of these social connections shape the tempo of cumulative cultural evolution.

Network architecture—here defined as a social structure with a characteristic set of properties—can shape transmission of behaviours, thus setting the tempo of CCE—here defined as the rate of cultural recombination events. For example, architectures with low network connectivity (i.e. density; the proportion of realized connections), high clustering (tendency of connected individuals to share the same social neighbours) and high modularity (tendency of the network to contain sets of individuals more connected to each other than with others) slow down the spread of information across populations [16–18]. The slower spread can then potentially favour greater cultural diversity by allowing multiple cultural lineages to arise in populations before any one lineage dominates [19,20]. While previous work has largely focused on how new behaviours spread through a social network [16,17] to establish cultures [20,21], and how cultural traits can generate a feedback shaping network structure [18,22], more recently it has been argued that emergent network properties could affect CCE [12,14,19] by shaping how new traits are produced, recombined and maintained [14]. For example, partial connectivity facilitates the emergence of multiple cultural lineages in parallel [20], which is required for achieving cultural accumulation, but partially connected networks suffer from cultural loss if connectivity is too low for new innovations to spread [14]. By contrast, full connectivity facilitates the rapid spread of new innovations, but can prevent the accumulation of alternative cultural traits [12,14]. However, within a given level of connectivity, how connections are structured—the social network architecture—could also impact CCE by influencing how fast and widely information can spread.

Because network architecture can shape the effect of connectivity on diffusion dynamics [23], those architectures that balance the ability for cultural accumulation together with the recombination of different cultural traits should have a selective advantage in facilitating CCE [19]. Multilevel societies, such as those in modern hunter–gatherers, feature high clustering and nested modularity. These network properties are expected to favour CCE by allowing coexistence of multiple cultural traits in different parts of the network, and for different cultural lineages to come into contact to allow combinations from lineages to produce new traits [19]. Multilevel societies have been demonstrated to accelerate CCE when compared to fully connected networks [19]. However, when considering their potential for facilitating CCE, multilevel and fully connected networks represent possible endpoints along a continuum of possible architectures. Here, we ask how a range of social network architectures can affect the tempo of CCE within a given population size and number, or density, of social connections within that population. Our approach allows us to explicitly disentangle the relative contribution of network architecture from those of connectivity and population size.

## 2. Material and methods

### (a) Overview

We first generated social networks with six different architectures—random, small-world, lattice, modular, modular lattice and multilevel—capturing different levels and combinations of clustering and modularity (figure 1a). We expressed these network architectures in populations with different sizes and densities of

connections (average degree), where all individuals in the network had the same degree. We then built two agent-based models to explore how network architecture affects cumulative culture evolution. Briefly, our models allow innovations of cultural products to take place along two cultural lineages, with the knowledge of new products being spread through social connections via two transmission mechanics: either one-to-many or one-to-one diffusion. Once a high level of product diversity has been reached in both lineages, agents can recombine each lineage's products into one with a final higher-payoff product (hereafter 'recombination'). Finally, we compared the performance of agents arranged in the different network architecture in terms of time to cultural recombination (i.e. tempo), time to diffusion and the diversity of cultural traits.

### (b) Social network architectures

We generated networks in which nodes representing individuals were linked by binary social relationships to represent the following six stereotypical social structures. We generated: (i) small-world networks, using the Watts–Strogatz model [23] with node degree $K$ links; (ii) random networks, by randomly connecting nodes ensuring all nodes had the same degree $K$; (iii) lattices, by placing nodes on a grid and connecting each to its $K$ nearest neighbours; (iv) modular networks, by assigning nodes into nine modules, randomly connecting each to $K-1$ nodes from the same module and one node from another module; (v) modular lattices, as per modular networks, but where the connections within modules were lattices; and (vi) multilevel networks, as per modular lattices, but assigning nodes into three sets of three modules, and connecting each to $K-2$ nodes within their module, one node from each module from within its set and one node from a module outside of it.

Each network architecture differs in clustering and modularity, but we standardize their connectivity (degree) and population size. For each architecture, we generated an ensemble of networks with different sizes ($N \in \{64,144,324\}$ nodes), and densities of connections (in which $K \in \{8,12,18,24,30\}$ average links). We used these population sizes because they allowed us to partition the network into equally sized clusters composed of equally sized groups in which all individuals had the same degree, and in which connectivity was greater within groups and within clusters than between groups and between clusters. Although all networks had comparable sizes and densities (i.e. the proportion links), the six architectures varied in levels and combinations of clustering coefficient and modularity. Clustering, $C$, informs the tendency of connected nodes to share the same connections with other nodes, while modularity, $Q$, informs the tendency of the nodes to be organized into cohesive subsets that are more connected to each other than to the rest of the network [24].

### (c) Cultural evolution simulations

We constructed two agent-based models to simulate cultural evolution on the different types of network architectures. The two models differed in the mechanics of information transmission: one-to-many versus one-to-one diffusion pathways. Our first agent-based model (model 1) followed Migliano et al. [19]. All agents were initialized with an inventory of three items from each of two lineages. In each simulation round (epoch), each focal agent was selected once, at random, and a partner randomly chosen from its social network connections. These agents combined one or two items from their inventory in proportion to their value into a triad of items. If this triad was a valid product, knowledge of that product was learned, spread immediately to all their network connections (one-to-many diffusion), and subsequently became available as an ingredient for making new products. Simulations finished once

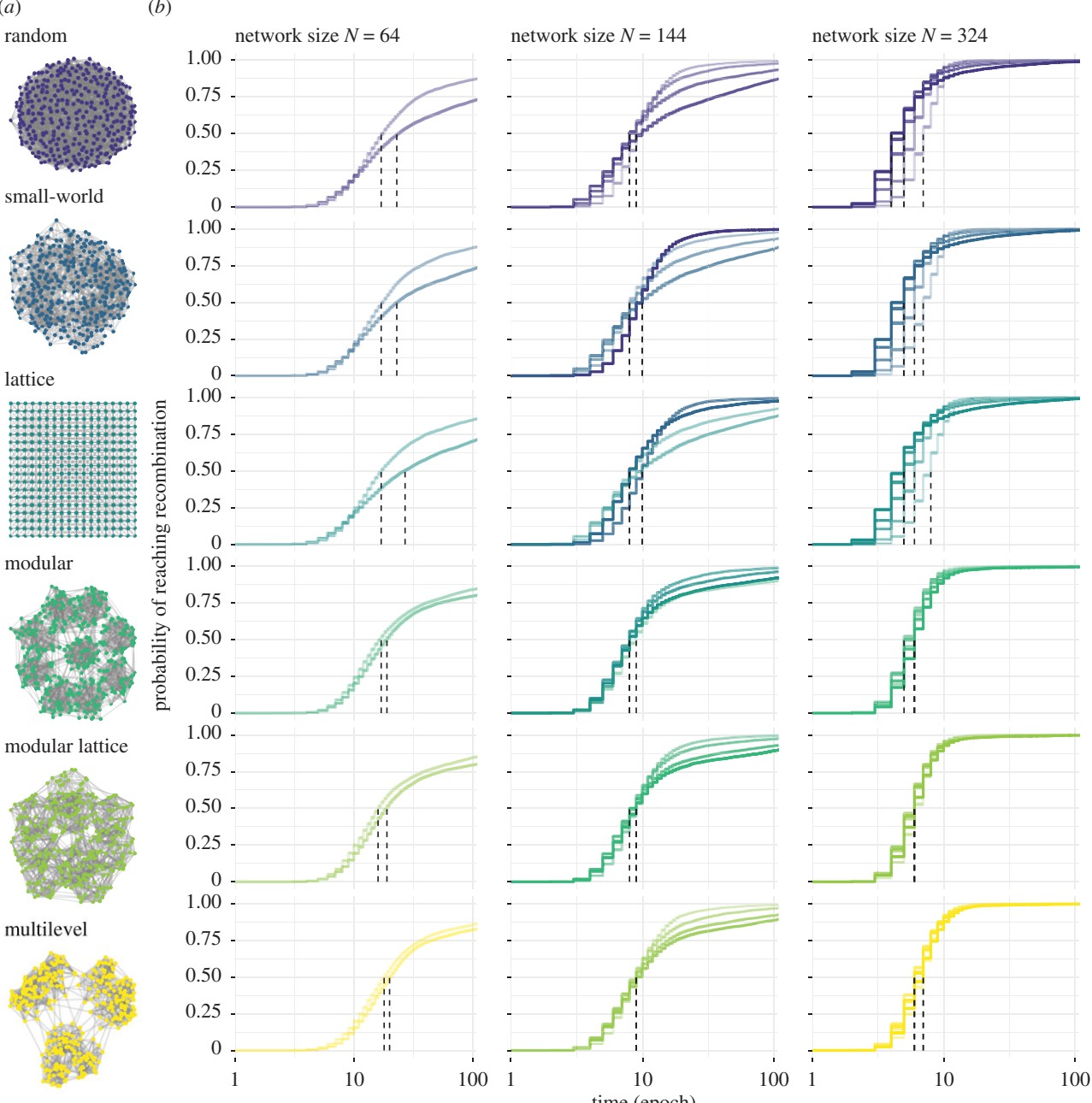

**Figure 1.** Social network architectures, and the time to recombination for each architecture across population sizes and levels of connectivity using model 1. (*a*) Network architectures vary in clustering and modularity: Random (unclustered $C = 0.03$, non-modular $Q = 0.24$), small-world (clustered $C = 0.52$, medium-modular $Q = 0.63$), lattice (clustered $C = 0.45$, medium-modular $Q = 0.54$), modular (unclustered $C = 0.23$, modular $Q = 0.82$), modular lattices (clustered $C = 0.41$, modular $Q = 0.81$), multilevel (clustered $C = 0.42$, modular $Q = 0.83$). Each binary network depicts populations with the same number of individuals (here, $N = 324$ nodes) that have the same number of social connections (here, degree $K = 12$ links per node; density $D = 0.037$) but are wired differently. (*b*) Cumulative incidence of recombination events (*y*-axis) as a stepwise function over time (*x*-axis, log epochs) for small ($N = 64$), medium ($N = 144$) and large population sizes ($N = 324$). The line shading represents the amount of network connectivity (node degree $K$, where the lighter the shade, the smaller the degree ($K \in \{8,12\}$ for $N = 64$; $K \in \{8,12,18,24\}$ for $N = 144$; $K \in \{8,12,18,24,30\}$ for $N = 324$). Vertical dashed lines indicate the median of time to recombination ($S(t) \leq 0.5$) per network connectivity, across architectures. The time to reach recombination was truncated to 100 epochs for better visualization. Curves were calculated based on 5000 simulations. (Online version in colour.)

a recombination product (a triad that recombines specific products from both lineages) was first innovated. We ran 5000 simulations for each of the network architecture types, sizes and densities of connections, recording time to achieve the recombination product (in epochs) and tracking the diversity of cultural innovations over time. An epoch was one simulation round in which each agent was selected once as a focal agent in random order.

Our second agent-based model (model 2) extended the first by changing the transmission mechanic and altering the set of valid combinations such that the model can run past the first

innovation of either recombination product. Transmission of valid products now occurred between dyads of agents (one-to-one diffusion) prior to choosing items from their inventory, in contrast to the broadcast style of diffusion in model 1. Secondly, if a triad contained either recombination products, the final product was that recombination product. In the case where both recombination products were present in the triad, one was chosen as the final product at random. This allowed us to track the diffusion of recombination products beyond their innovation. We also ran 5000 simulations for the same parameter space of model 1, recording time (in epochs) to cultural lineage

recombination, as well as time to diffusion to the majority of the network (i.e. the latency to more than $(N/2)+1$ nodes having a high-value product).

## (d) Data analyses

We compared the performance of agents organized in the different social network architectures in terms of the time to recombination, time to diffusion and the diversity of cultural lineages over time. To compare time to recombination, we used time-to-event (survival) analyses [25] where time to recombination was a function of network architecture and connectivity. Our simulations yielded time-to-event data, so we used the following methods from survival analysis, which are suitable for such data. For each population size, we used the cumulative incidence function to estimate the proportion of simulations in which agents reached the recombination of each cultural lineage's products into a final high-payoff product (the 'event'). We used the non-parametric Kaplan–Meier product limit estimator to estimate the 'survival function' from this time-to-event data; since we represented the time intervals based on observed recombination events from 5000 simulations from model 1, we also calculated the 95% confidence intervals (using the Greenwood estimator). To measure variance in time to recombination across population sizes, we measured the quartile coefficient of dispersion ($QCD = (Q3 - Q1)/(Q3 + Q1)$), as this variable is not normally distributed and QCD offers a more robust measure.

While statistics are not typically performed on data from agent-based models since the posterior is directly sampled, we wanted to quantify the relative contributions of architecture, size and connectivity without the cumbersome descriptions of the entire distribution (distributions can be readily seen in figure 2). For both models 1 and 2, we created three sets of generalized linear models (GLMs) that predicted logged time to recombination (in epochs). Time to recombination was logged to account for non-normality of residuals, and to make comparisons more fair by bringing the mean closer to the median of the distribution. All models used log link function, as the data was nonlinear conditional on predictors, even after the log transformation. Also, the log link function allowed the presentation of exponentiated coefficients, which simplify the comparison to the reference (here, the random networks at the GLM intercept). The first set of GLMs used a full interaction structure to partition the relative contributions of architecture, size and connectivity to the average time to recombination (electronic supplementary material, table S1), excluding fully connected networks. To then compare fully connected networks to all other networks, we built a GLM using architecture and population size as predictors in a full interaction structure (electronic supplementary material, table S2). Connectivity was excluded as a predictor, as all fully connected networks only have 1 possible degree ($K = N - 1$). Finally, to compare differences between architectures more precisely, we subset data by connectivity and population size and performed a GLM with only architecture as a predictor for each subset, again excluding fully connected networks. We performed all data analyses in R [26], using 'survival' [27] and 'survminer' [28] packages.

## 3. Results

When comparing the time to cultural recombination across population sizes and levels of connectivity, partially connected networks consistently outperform fully connected networks [12,14,19]. A GLM indicated that fully connected networks were, on average, 65% slower (GLM, $\exp(\beta) = 1.652$, $t = 59.208$, $p < 0.001$; electronic supplementary material, table S1) compared to the least structured network architecture of the same size (random, $N = 64$ taken as the intercept), with similar decreases in performance independent of size. Further, we also confirm [7] that larger populations take less time (about 40% less) to reach recombination (GLM, $\exp(\beta) = 0.618$, $t = -68.481$, $p < 0.001$) compared to networks of the same architecture and connectivity (figures 1b and 2; electronic supplementary material, table S2). Larger partially connected network architectures were less variable in their time to recombination (quartile coefficient of dispersion: $QCD = 0.688$ for $N = 64$; $QCD = 0.444$ for $N = 144$; $QCD = 0.273$ for $N = 324$; figure 2a). We also found that time to recombination was optimized at intermediate densities of connections, confirming that intermediate levels of connectivity can favour CCE [14], and revealing that the optimal level of connectivity varied with population size (figure 1b). In the smallest population ($N = 64$), sparse networks outperformed the others, but this was reversed in the largest population ($N = 328$) (figure 1b). However, differences in time to recombination were generally small (figures 1b and 2a).

## (a) Architectures favouring cumulative cultural evolution under some conditions disfavour it under other conditions

Under the one-to-many diffusion mechanism (model 1), multilevel, modular and modular lattice architectures had relatively shorter times to recombination in smaller populations with greater connectivity and in larger populations with less connectivity (figure 3a). However, such network architectures performed worse than lattice, small world and random architectures in smaller populations with less connectivity and in larger populations with greater connectivity (figure 3a). Multilevel, modular and modular lattice architectures were optimal at lower and higher levels of connectivity in medium-sized populations ($N = 144$, figure 3a), although connectivity generally had a lesser impact for these architectures relative to random, small-world and lattice architectures (figure 1b). Surprisingly, multilevel performed the worst in seven out of the 11 size and connectivity combinations (figure 3a) despite having the highest clustering and modularity—properties that have been predicted to favour CCE [19]. The modular lattice architecture (which had similar modularity and clustering to multilevel architecture) performed best in the other four combinations (figure 3a). Thus, no one network architecture proved optimal, with those favoured under some conditions being disfavoured under other conditions.

The large variation in the time to recombination (figure 2) within a given combination of network architecture, population size and density of connections suggests that the outcomes of a simulation were predominantly driven by stochastic events. The impact of such stochasticity is best revealed by the bimodal outcome for partially connected networks, which arises most often in smaller populations (figure 2a). This bimodality occurs because there are fewer independent innovation events when there are fewer individuals, which increases the chance that cultural products all emerge from the same lineage and, therefore, that this single lineage spreads to the whole population before the other lineage is innovated. Tracking the diversity of products over time (figure 4; electronic supplementary material, figure S1) highlights how the stochasticity in early events can affect cultural diversity, and therefore the outcomes of CCE, even within

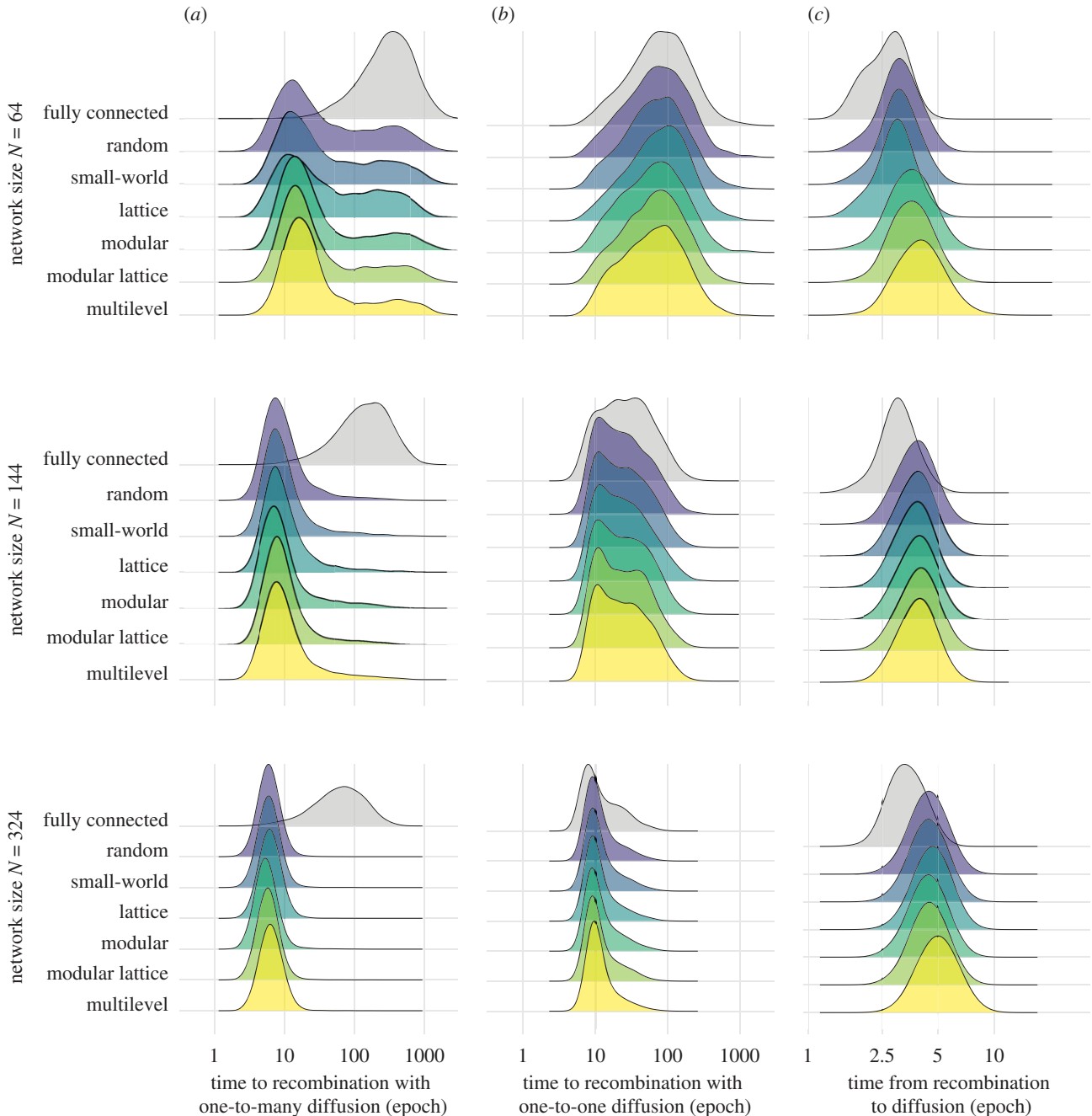

**Figure 2.** Time to recombination and time from recombination to diffusion across network architectures with varying sizes but a fixed degree. Comparison of the performance across the range of network architectures of the same degree (here $K = 12$ links per node) and fully connected networks of the same size ($N = 64$, $N = 144$ and $N = 324$ nodes, $K = 63$, 143 and 323, respectively). (a) Time to recombination (log epochs) from 5000 simulations with model 1 that uses a broadcast (one-to-many) diffusion dynamic. (b) Time to recombination (log epochs) from 5000 simulations with model 2 that uses a dyadic (one-to-one) diffusion dynamic. (c) Difference between the time to recombination and the time to diffusion, where time to diffusion corresponds to the latency until the majority of the individuals in the population has information about the final higher-payoff product, from 5000 simulations using model 2 (one-to-one diffusion). All ridges were plotted with the same bandwidth (0.18). (Online version in colour.)

the same network architecture. Overall, measuring the tempo of CCE under one-to-many diffusion (model 1) revealed differences in the best performing architecture across population sizes and levels of connectivity (figures 2a and 3a); however, these between-architecture differences were small (range = 0.886–1.145; figure 3a), compared to the large variance in the time to recombination within architecture (figure 2a).

## (b) The same architectures that promote diversity also restrict transmissions of novel products

To identify the relative contribution of diffusion mechanisms to CCE, we extended model 1 by implementing a one-to-one

diffusion mechanism (model 2). Whereas model 1 represents an extreme scenario where information spreads instantaneously to all the contacts of a focal agent, model 2 tests another extreme in which information about discoveries spread only to a single contact at a time. Interestingly, when employing such one-to-one diffusion dynamics, fully connected networks were only estimated to be 3% slower to recombination compared to networks of the same population size (GLM, $\exp(\beta) = 1.033$, $t = 6.808$, $p < 0.001$; electronic supplementary material, table S1). Again, larger populations had a significantly shorter average time to recombination compared to smaller networks of the same architecture and degree (GLM, $\exp(\beta) = 0.595$, $t = -109.094$, $p < 0.001$; figure 2b,

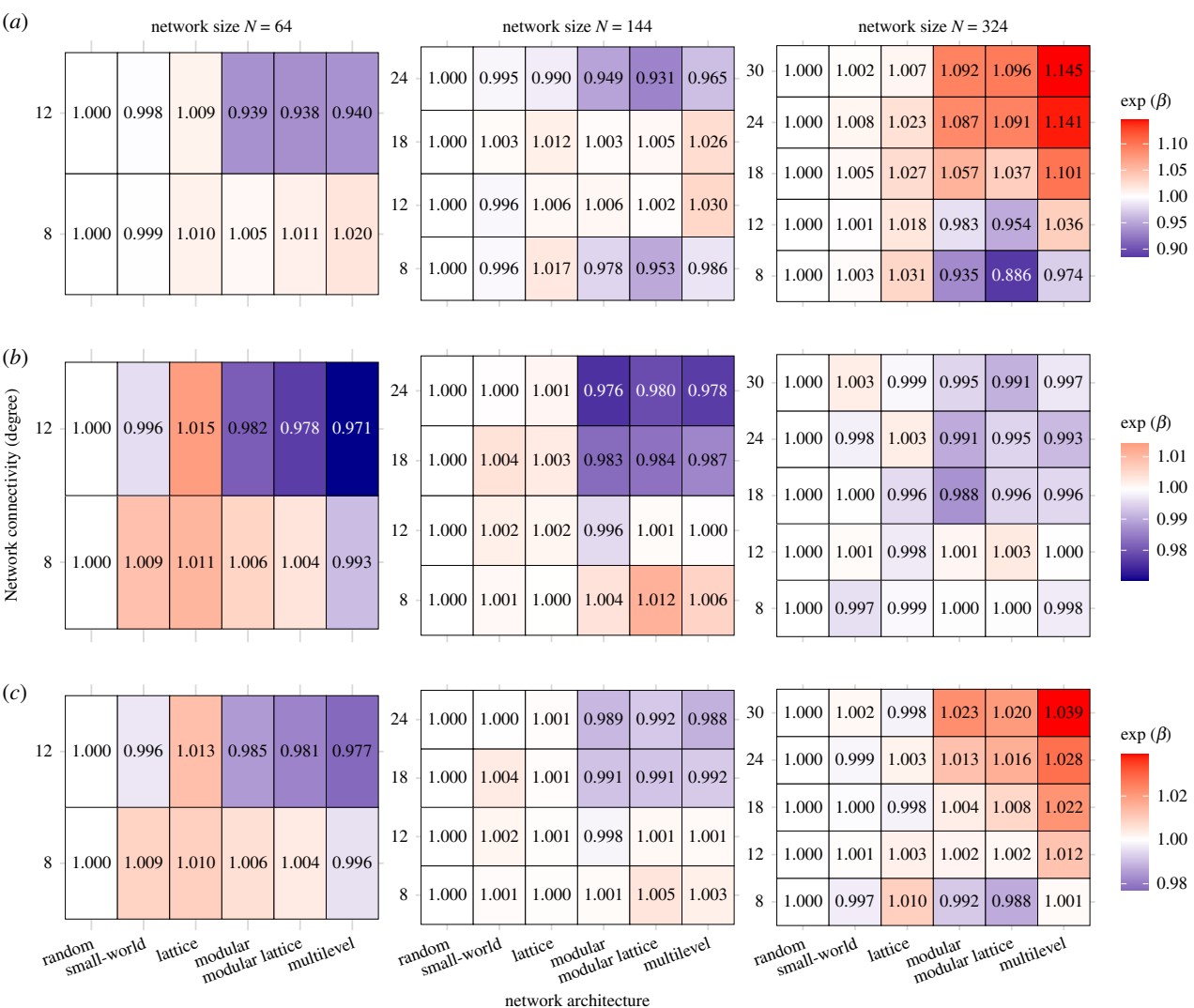

**Figure 3.** Relative performance of network architectures within each of the 11 combinations of population size and level of connectivity used in the simulations. Each row of each table reports the coefficient estimate of GLMs of network architecture (column) in function of the time to recombination while maintaining degree (row) and network size (box) constant. Higher coefficients (red colours) represent a poorer performance (longer latency to recombination) while lower coefficients (blue colours) represent architecture that perform better (shorter latency to recombination) for that combination of population size and level of connectivity (using random networks as the reference architecture in the GLM intercept). The relative performance of each architecture is shown for (a) time to recombination under a one-to-many diffusion mechanism (model 1), (b) time to recombination under a one-to-one diffusion mechanism (model 2) and (c) total time to diffusion (from simulation start until the majority of the population has information about the final higher-payoff product) under a one-to-one diffusion mechanism (model 2). (Online version in colour.)

electronic supplementary material, table S2). However, under one-to-one diffusion, the relative times to recombination of different architectures was generally more consistent than under one-to-many diffusion, both in their median times to recombination (figure 2b) and in their relative performance under a given population size and level of connectivity (figure 3b). Within a given population size, multilevel architecture typically had the shortest times to recombination when networks had greater connectivity, but there was almost no difference in performance among architectures when connectivity was low (figure 3b). Thus, in contrast to one-to-many, one-to-one diffusion increased the tempo for architectures with greater modularity and clustering (modular, modular lattice, multilevel) relative to the other architectures.

Model 2 also tracked the time for the recombination product to diffuse to the majority of the population, something which model 1 was not designed to track. The time from recombination to diffusion was shortest in fully connected networks, and increased with population size (figure 2c). When evaluating performance from the start of the simulations until the time to diffusion, population size caused

the most variation in outcomes, compared to network architecture or connectivity (electronic supplementary material, table S2). In small populations, the contribution of the final diffusion was relatively small compared to the time to recombination, meaning that the best performing networks in achieving recombination also performed best overall (figure 3c). By contrast, in larger populations, the performance of modular and clustered network architectures (modular, modular lattice and multilevel) all performed worse: they were the slowest at reaching final diffusion (figure 3c) despite typically reaching recombination the fastest (figure 3b). These differences, however, remain minor relative to the variance in outcomes within each set of conditions (architecture, population size and connectivity).

## 4. Discussion

We revisited recent empirical and *in silico* experiments in humans to tease apart the contributions of different candidate social structures to the tempo of cumulative cultural

**Figure 4.** Cultural product diversity across time in a fully connected social network and in a highly structured network architecture. The distributions of time to recombination and cultural product diversity illustrates how early stochasticity can affect cultural diversity, even within the same network architecture, and therefore, can shape the outcomes of cumulative cultural evolution. (a) Time to recombination (epochs) from 5000 simulations with one-to-many diffusion dynamics (model 1) in multilevel and fully connected networks (with $N = 64$ and $K = 12$) highlights distinct cultural trajectories among the highly structured networks (note the bimodal distribution). Following panels show cultural diversity over time from one simulation taken from the (b) single mode of the fully connected network, and the (c) first (*) and (d) second (**) modes of the distribution of results from the multilevel architecture. Cultural diversity (y-axis) represents the proportion of the population with one of the possible products over time: a combination of two inventory items (2nd stage; thin full lines), a valid combination triad of items (3rd stage; dashed lines) and the final higher-payoff product, i.e. a triad recombining products from the two lineages (recombination; thick full lines). These products could come from two independent cultural lineages: lineage A (top row) and lineage B. For better visualization, the distribution of single inventory items (1st stage product) was omitted (but see electronic supplementary material, figure S1). (Online version in colour.)

evolution. Our results suggest that it is unlikely that one specific social network architecture consistently promotes cumulative cultural evolution across all population sizes, densities of social connections or diffusion mechanisms. Rather, the relationship is nuanced; the broad distribution of outcomes from our two models indicate that the best performing architecture under some conditions can be the worst performing architecture under others. Further, the outcome of any diffusion mechanism is as likely to be affected by stochastic processes as by the architecture of the networks itself. While not at odds with previous work showing that multilevel societies can accelerate cumulative cultural evolution [19], our results suggest that a range of other partially connected architectures could equally increase the tempo of cumulative cultural evolution.

The fact that alternative architectures can have similar outcomes in terms of CCE has important consequences for how the social structure of societies and CCE are framed in future discussions, and where future research is directed. Current thinking is that complex, highly structured societies, such as multilevel societies, might precede recombinatory CCE in the timeline of human evolution, or that the benefits accrued from cultural evolution [22] or CCE [19] might co-evolve with clustered and modular network structures. However, our results suggest that simple patterns of spatial distribution (e.g. a lattice social network caused by distributed resources) could lead to largely equivalent effects on CCE. It follows that we might expect to find recombinatory CCE even before the evolution of complex societies. Indeed, evidence that simple, lattice-like social structures [29] can provide a substrate for recombinatorial culture might be provided by the combinatorial, spatially variable song structure of territorial passerine birds [30–34], which several authors have proposed to be a simple form of CCE [35,36].

Population size has been suggested as another major demographic factor affecting rates of CCE [3,7–9]. Our findings align with this previous research, with our simulations showing that larger populations always have a higher rate of cultural accumulation. Population size also interacted with connectivity (which we modelled as a fixed network degree, i.e. the number of individuals' social connections [22]), with changes in connectivity having a more pronounced effect in smaller populations. This outcome is likely to arise because an increase in one unit of mean degree corresponds to a greater increase in network connectivity in smaller populations (more rapidly pushing the network towards becoming fully connected). However, in our simulations, we did not vary the distribution in connectivity among individuals, which has previously been shown to impact the properties of information cascades [37] and differences among groups in behaviours such as cooperation [38]. Skewed degree distributions, where some nodes are much more connected than others, could allow independent lineages to arise in peripheral nodes and for highly connected 'hubs' to combine the products from these lineages, thereby facilitating CCE. Thus, variation in how much or how little individuals are connected, independently of other factors (mean connectivity, population size and network properties), is an important dimension for future studies on CCE to consider.

Fully connected networks have been commonly used to evaluate the performance of a transmission network with a given set of characteristics [3,19]. However, a fully connected social network representing a human population of any reasonable size would represent an unrealistically high level of connectivity [39], even in the fractal-like human social networks [40]. In addition, our simulations demonstrate that the contribution of large differences in connectivity to the rates of CCE outweighs any effects of network architecture, at least when information is broadcast (i.e. a one-to-many diffusion mechanism). Thus, we suggest that fully connected networks are uninformative null models for testing the influence of social structure on the tempo of CCE. Instead, random networks of similar sizes and densities of connections as the network of interest would provide a more robust benchmark (see also [41]). Our results suggest, however, that any effect of

network architecture on increased rate of CCE inferred from noisy field data would probably be indistinguishable from the null expectancy, as variation within architecture greatly exceeded that of between architectures.

One major remaining question is how the diversity of cultural lineages might affect CCE across different network architectures. Our simulations were built on previous empirical and theoretical work that considered two cultural lineages [12,19]. Increasing the number of lineages could potentially reduce or increase the time to recombination, depending on how the recombinations have to be made and on their payoffs. For example, recombinations from any two of three or more lineages would probably emerge faster, as the probability that any two lineages survive is higher. By contrast, recombinations requiring products from three lineages could take substantially longer, especially in more connected networks (or with one-to-many diffusion dynamics), since all three would have to become established. However, adding more lineages would be unlikely to reduce the influence of stochasticity in any given diffusion. This is especially the case in highly connected networks or one-to-many diffusion dynamics, because it would not stop one lineage from initially dominating (when a single invention spreads throughout the population, before another lineage is invented; see figure 4). Rather, the tendency for one lineage to dominate would be more heavily impacted by the payoff structure associated with incremental improvements to products. While we do not believe that increasing the number of lineages would substantially change the primary findings of this study, a promising avenue for further research would be a more in-depth exploration of how exposure to multiple cultural lineages may shape the tempo and mode of CCE.

The evolutionary benefits of CCE not only rely on cultural accumulation, but also on the ability for new cultural traits to spread through populations. When we extended simulations to examine the time from recombination to the diffusion of the final higher-payoff product, our results suggested that the network architecture hypothesized to improve time to recombination (multilevel) paradoxically inhibited diffusion most. These findings complement and extend previous studies demonstrating that populations with partially connected network structures can suffer from cultural loss when connectivity becomes too low for new innovations to spread [12], and that cultural evolution is more profoundly impacted by the rates of information loss and transmission than differences in social network architecture [41]. Further, we show that the relative performance of network architectures can change dramatically when considering performance in terms of acquisition of behaviour by the majority of individuals in a population, as opposed to the time when a single individual has reached cultural recombination, especially in larger populations. For example, while populations in a multilevel social network architecture consistently reached recombination faster than those organized as random networks under one-to-one diffusion, the multilevel architecture then restricted the final spread of higher-value cultural traits. These results therefore suggest that multiple dimensions of performance—including every step from innovations to the final acquisition of higher-valued traits—may need to be considered when studying the role of social structure in shaping CCE and vice versa.

Our work reinforces the need for studies of CCE to explicitly consider how network structure interacts with transmission mechanisms to form a realized transmission network. We show that a very restrictive transmission dynamic (one-to-one) mitigates the effect of network connectivity on CCE by generating a partially connected transmission network within an otherwise fully connected social network. The consequences of transmission dynamics on CCE were demonstrated, for instance, by Migliano et al. [19] who found that CCE was faster in simulations where transmission was limited to kin-based connections (i.e. reduced connectivity). Under one-to-one diffusion, independent lineages can develop in fully connected networks because new information is not immediately accessible to all, leading to more comparable performance between fully and partially connected networks. Thus, when simulating CCE, it is important to match the transmission dynamics with the time scale of the model. One-to-many diffusion can be realistic when each epoch represents one generation (e.g. the innovation of a new medicine [19] could take tens or hundreds of epochs to reach high recombinatory levels), while the one-to-one diffusion might be more realistic when cultural traits are simpler to recombine. The production and innovation frequency, as well as transmission biases, may further vary between species, populations, tasks and contexts. Together with network structure, innovation frequency and transmission biases may fundamentally alter the transmission dynamics—for example, conformity overrides payoff biases [21,42] and homophily reduces social connectivity [18,43]—fuelling evolutionary feedbacks between network structure and cultural evolution [22]. Both factors will therefore alter the resulting transmission networks, potentially restricting the spread of new cultural traits and slowing recombinatory CCE. More than highlighting the intricate, yet nuanced, interplay between demography and cultural transmission, our work strengthens our emerging understanding that realized connectivity, rather than network architecture, is important for cumulative cultural evolution [2,3].

Data accessibility. We implemented the agent-based models in R and Python. The code to generate the social networks, perform the simulations and replicate the data, the statistical analyses, and the figures are available at https://github.com/simeonqs/Social_network_archi-tecture_and_the_tempo_of_cumulative_cultural_evolution [44]. The simulated data used in this manuscript are available from the Dryad Digital Repository: https://dx.doi.org/10.5061/dryad.3r2280gff [45].

Authors' contributions. D.R.F., D.P. conceived the idea. M.Ch., S.Q.S., D.R.F. developed and ran the models and simulations. M.Ca., M.Ch. analysed the data. M.Ca., L.M.A., D.R.F., M.Ch. drafted the initial manuscripts, and all authors contributed to writing and editing the final article.

Competing interests. We declare we have no competing interests.

Funding. This work was supported by the Max Planck Society, and the Advanced Centre for Collective Behaviour, funded by the Deutsche Forschungsgemeinschaft (DFG, German Research Foundation) under Germany's Excellence Strategy (EXC 2117–422037984). D.R.F. and D.P. were funded by a grant from the European Research Council (ERC) under the European Union's Horizon 2020 research and innovation programme (grant agreement no. 850859), awarded to D.R.F. D.R.F. was also funded by an Eccellenza Professorship Grant of the Swiss National Science Federation (grant no. PCEFP3_187058). L.M.A. was funded by a Max Planck Independent Group Leader Fellowship. P.H. was funded by a scholarship from the China Scholarship Council (grant no. 201706100183). M.Ca. was funded by CAPES (88881.170254/2018-01) and the Max Planck Society. M.Ch., D.P. and S.Q.S. received funding from the IMPRS for Organismal Biology. Open Access funding provided by the Max Planck Society.

Acknowledgements. We thank the Social Evolutionary Ecology Lab and the Cognitive and Cultural Ecology Lab at the Max Planck Institute of Animal Behavior for the discussion that inspired this manuscript, and to the anonymous reviewers for the insightful comments.

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
