## [Peer Review File · Proceedings of the Royal Society B: Biological Sciences]

Review History

RSPB-2020-3107.R0 (Original submission)

Review form: Reviewer 1 (Marco Smolla)

Recommendation

Accept with minor revision (please list in comments)

Scientific importance: Is the manuscript an original and important contribution to its field?

Excellent

General interest: Is the paper of sufficient general interest?

Good

Quality of the paper: Is the overall quality of the paper suitable?

Excellent

Is the length of the paper justified?

Yes

Should the paper be seen by a specialist statistical reviewer?

No

Do you have any concerns about statistical analyses in this paper? If so, please specify them explicitly in your report.

No

It is a condition of publication that authors make their supporting data, code and materials available - either as supplementary material or hosted in an external repository. Please rate, if applicable, the supporting data on the following criteria.

Is it accessible?

Yes

Is it clear?

Yes

Is it adequate?

Yes

Do you have any ethical concerns with this paper?

No

Comments to the Author

From the outset of cultural evolution research understanding the transmission of information within groups and populations has played a major role. In recent years, observational and theoretical work has shown the importance of different aspects of this complex process on cultural dynamics, from transmission modes to population size and connectivity. Only recently have actual social network aspects moved into the field's focus. The present manuscript provides a systematic study of different network types, network sizes, and network connectivity, and their combined and individual effects on the emergence of new traits and their spread in a population.

I enjoyed reading this manuscript. In my opinion, the topic is timely and relevant to the field. There is very little to criticise about the overall presentation of the work both in text and figures. The Background provides relevant information. The methods are (mostly) written clear and concise. The results are generally well presented and discussed.

Here is a list of specific comments:

II. 163-168 "For each population size, we used the Cumulative Incidence Function to estimate the proportion of simulations in which agents reached the recombination of each cultural lineage's products into a final high-payoff product. We used the nonparametric Kaplan-Meier product limit estimator to represent the time intervals based on observed recombination events from 5,000 simulations from model 1, calculating 95% confidence intervals with the Greenwood estimator." - Compared to the otherwise detailed description in the methods, there are three concepts mentioned here that should probably receive a little bit more attention. Why do the authors use them, how do they work, and/or what do they mean?

I. 270 "Fig. S5" - There is neither a Figure 5 nor a Figure S5 in the files I could access

II. 322-324 "Our simulations demonstrate that the contribution of large differences in connectivity outweighs any effects pertaining to architecture, at least when information is broadcast (i.e. a one-to-many diffusion mechanism)." - Maybe rephrase. What do the authors mean by the contribution of large differences in connectivity?

Figure 4 was confusing to me. What are the y-axes showing? In B-D there are too many lines colours and shades, the message is not clear, it is neither intuitive to grasp what the authors want to convey here, nor is it covered by the figure's caption.

Figure S1, the different types of lines are hard to differentiate, maybe consider using different colours for the four possible traits (instead of using it for the different network types, which are already clearly delineated by the graph plots and the headings).

The deposited data, code, and results on GitHub are a great addition to the manuscript. (Note: I noticed that some files are missing, e.g. output for model 1 and model 2 in 3_R_agent_based_models, and the code for figures 1,2, and 4 in 4_create_figures). If possible, and to improve replicability, I would suggest adding even more comments, especially to the simulation code.

Review form: Reviewer 2

Recommendation

Accept with minor revision (please list in comments)

Scientific importance: Is the manuscript an original and important contribution to its field?

Good

General interest: Is the paper of sufficient general interest?

Excellent

Quality of the paper: Is the overall quality of the paper suitable?

Good

Is the length of the paper justified?

Yes

Should the paper be seen by a specialist statistical reviewer?

Yes

Do you have any concerns about statistical analyses in this paper? If so, please specify them explicitly in your report.

No

It is a condition of publication that authors make their supporting data, code and materials available - either as supplementary material or hosted in an external repository. Please rate, if applicable, the supporting data on the following criteria.

Is it accessible?

Yes

Is it clear?

Yes

Is it adequate?

Yes

Do you have any ethical concerns with this paper?

No

Comments to the Author

This paper presents two agent-based models that explore how network architecture affects cumulative culture evolution. Six different network architectures (random, small-world, lattice,

modular, modular lattice, and multilevel) capturing different levels and combinations of clustering and modularity have been implemented in populations with different sizes and densities of connections.

The simulations confirm recent results that combinatorial innovation is optimized at intermediate densities of connections but also reveals that the optimal level of connectivity varies with population size. Although not entirely surprising, this result is important to the current debate regarding whether humans' unique multilevel social structure accelerates cumulative cultural evolution.

The comparison between the two models implementing either one-to-many or one-to-one diffusion processes also reveals interesting results and highlight the complex interaction between population size, structure and transmission mechanisms.

My opinion is that, although the paper does not yield any major finding, it provides a lot of food for thought to the large community of scholars interested in the relationship between demography and cumulative culture. The goal of the paper of disentangling the relative contribution of network architecture from those of connectivity and population size is clearly an important one.

My only concern is that the authors limited their investigations to a situation where innovations of cultural products can only take place along two cultural lineages. I don't think this threatens the validity of their main result (that is optimal level of connectivity varies with population size) but this clearly increases the chance that cultural products all emerge from the same lineage. I would expect the results of their simulations to be less influenced by stochastic events were the fitness landscapes to include more cultural lineages. I would recommend the authors to either run additional simulations to verify this relationship or explicitly mention this limitation when they discuss the role of stochastic events on cumulative culture.

Decision letter (RSPB-2020-3107.R0)

25-Jan-2021

Dear Dr Cantor:

Your manuscript has now been peer reviewed and the reviews have been assessed by an Associate Editor. The reviewers' comments (not including confidential comments to the Editor) and the comments from the Associate Editor are included at the end of this email for your reference. As you will see, the reviewers and the Editors have raised some concerns with your manuscript and we would like to invite you to revise your manuscript to address them.

Research ethics:

Use of animals and field studies:

It is a condition of publication that you make available the data and research materials supporting the results in the article. Please see our Data Sharing Policies (<https://royalsociety.org/journals/authors/author-guidelines/#data>). Datasets should be deposited in an appropriate publicly available repository and details of the associated accession number, link or DOI to the datasets must be included in the Data Accessibility section of the article (<https://royalsociety.org/journals/ethics-policies/data-sharing-mining/>). Reference(s) to datasets should also be included in the reference list of the article with DOIs (where available).

Please submit a copy of your revised paper within three weeks. If we do not hear from you within this time your manuscript will be rejected. If you are unable to meet this deadline please let us know as soon as possible, as we may be able to grant a short extension.

Best wishes,
Dr Robert Barton
mailto: proceedingsb@royalsociety.org

Associate Editor
Board Member: 1
Comments to Author:

This paper uses agent-based models to examine how the structure of social networks influences cumulative cultural evolution. This is a critical issue in the field of cultural evolution, with relevance across a range of fields and the two reviewers agree, as do I, that the paper is interesting and timely. However, as Reviewer 1 points out, there are a number of places where revisions are needed to provide further detail and clarity on aspects of the methodology and presentation of results (some figures also seem to be missing). Reviewer 2 has a deeper concern that, by allowing innovations to occur along only two cultural lineages the models artificially increase the probability that all cultural products emerge from the same lineage. If possible, additional simulations would be very useful to address this concern.

Reviewer(s)' Comments to Author:

Referee: 1

Comments to the Author(s)

From the outset of cultural evolution research understanding the transmission of information within groups and populations has played a major role. In recent years, observational and theoretical work has shown the importance of different aspects of this complex process on cultural dynamics, from transmission modes to population size and connectivity. Only recently have actual social network aspects moved into the field's focus. The present manuscript provides a systematic study of different network types, network sizes, and network connectivity, and their combined and individual effects on the emergence of new traits and their spread in a population.

I enjoyed reading this manuscript. In my opinion, the topic is timely and relevant to the field. There is very little to criticise about the overall presentation of the work both in text and figures. The Background provides relevant information. The methods are (mostly) written clear and concise. The results are generally well presented and discussed.

Here is a list of specific comments:

II. 163-168 "For each population size, we used the Cumulative Incidence Function to estimate the proportion of simulations in which agents reached the recombination of each cultural lineage's products into a final high-payoff product. We used the nonparametric Kaplan-Meier product limit estimator to represent the time intervals based on observed recombination events from 5,000 simulations from model 1, calculating 95% confidence intervals with the Greenwood estimator." - Compared to the otherwise detailed description in the methods, there are three concepts mentioned here that should probably receive a little bit more attention. Why do the authors use them, how do they work, and/or what do they mean?

I. 270 "Fig. S5" - There is neither a Figure 5 nor a Figure S5 in the files I could access

II. 322-324 "Our simulations demonstrate that the contribution of large differences in connectivity outweighs any effects pertaining to architecture, at least when information is broadcast (i.e. a

one-to-many diffusion mechanism)." – Maybe rephrase. What do the authors mean by the contribution of large differences in connectivity?

Figure 4 was confusing to me. What are the y-axes showing? In B-D there are too many lines colours and shades, the message is not clear, it is neither intuitive to grasp what the authors want to convey here, nor is it covered by the figure's caption.

Figure S1, the different types of lines are hard to differentiate, maybe consider using different colours for the four possible traits (instead of using it for the different network types, which are already clearly delineated by the graph plots and the headings).

The deposited data, code, and results on GitHub are a great addition to the manuscript. (Note: I noticed that some files are missing, e.g. output for model 1 and model 2 in 3_R_agent_based_models, and the code for figures 1,2, and 4 in 4_create_figures). If possible, and to improve replicability, I would suggest adding even more comments, especially to the simulation code.

Referee: 2

Comments to the Author(s)

This paper presents two agent-based models that explore how network architecture affects cumulative culture evolution. Six different network architectures (random, small-world, lattice, modular, modular lattice, and multilevel) capturing different levels and combinations of clustering and modularity have been implemented in populations with different sizes and densities of connections.

The simulations confirm recent results that combinatorial innovation is optimized at intermediate densities of connections but also reveals that the optimal level of connectivity varies with population size. Although not entirely surprising, this result is important to the current debate regarding whether humans' unique multilevel social structure accelerates cumulative cultural evolution.

The comparison between the two models implementing either one-to-many or one-to-one diffusion processes also reveals interesting results and highlight the complex interaction between population size, structure and transmission mechanisms.

My opinion is that, although the paper does not yield any major finding, it provides a lot of food for thought to the large community of scholars interested in the relationship between demography and cumulative culture. The goal of the paper of disentangling the relative contribution of network architecture from those of connectivity and population size is clearly an important one.

My only concern is that the authors limited their investigations to a situation where innovations of cultural products can only take place along two cultural lineages. I don't think this threatens the validity of their main result (that is optimal level of connectivity varies with population size) but this clearly increases the chance that cultural products all emerge from the same lineage. I would expect the results of their simulations to be less influenced by stochastic events were the fitness landscapes to include more cultural lineages. I would recommend the authors to either run additional simulations to verify this relationship or explicitly mention this limitation when they discuss the role of stochastic events on cumulative culture.

Author's Response to Decision Letter for (RSPB-2020-3107.R0)

See Appendix A.

Decision letter (RSPB-2020-3107.R1)

08-Feb-2021

Dear Dr Cantor

I am pleased to inform you that your manuscript entitled "Social network architecture and the tempo of cumulative cultural evolution" has been accepted for publication in Proceedings B.

Open Access

Paper charges

Sincerely,

Dr Robert Barton

Associate Editor:

Board Member

Comments to Author:

(There are no comments.)

Appendix A

Manuscript ID RSPB-2020-3107

Title: Social network architecture and the tempo of cumulative cultural evolution

Authors: M Cantor, M Chimento, SQ Smeele, P He, D Papageorgiou, LM Aplin, DR Farine

To the editorial board at the Proceedings of the Royal Society

Editor Robert Barton, PhD

Dear Dr. Barton,

Thank you for your interest on our work and for the welcoming editorial decision our manuscript RSPB-2020-3107.

We appreciate the time invested by the two Referees and the Associate Editor in this thorough review. We were particularly encouraged by the positive remarks by the Reviewer#2 highlighting that the *“goal of the paper of disentangling the relative contribution of network architecture from those of connectivity and population size is clearly an important one”* as well as by the Reviewer#1 who emphasize that our work *“is timely and relevant”*.

In addition to such very positive feedback, both Reviewers provided additional critical comments. We found these comments extremely helpful for improving the readability of our text and figures. We addressed every comment, giving special consideration to the suggestion by Reviewer #2 on the number of cultural lineages considered in our simulations. Reviewer #2 suggested that the influence of stochasticity might be lower when including more than two cultural lineages. We decided to address this in the discussion, since running simulations with more than two lineages would require multiple decisions about how to implement these (e.g. should the final high pay-off product include products of all lineages or only two?), which in turn could influence the outcomes to beyond the scope of our current manuscript. In our revised Discussion, we now discuss how future work can address these open questions appropriately.

By addressing these suggestions, we now feel that our work has gained both in value and clarity. Please find our point-by-point responses below, as well as the revised manuscript with all changes tracked. We therefore resubmit the revised version of our manuscript for your consideration for publication in Proceedings of the Royal Society.

Thank you for the insightful reviews and excellent editorial services provided.

Best wishes,

Mauricio Cantor, Michael Chimento, Simeon Smeele, Peng He, Danai Papageorgiou, Lucy Aplin, Damien Farine

COMMENTS BY THE ASSOCIATE EDITOR:

Comment#1 by Associate Editor: This paper uses agent-based models to examine how the structure of social networks influences cumulative cultural evolution. This is a critical issue in the field of cultural evolution, with relevance across a range of fields and the two reviewers agree, as do I, that the paper is interesting and timely. However, as Reviewer 1 points out, there are a number of places where revisions are needed to provide further detail and clarity on aspects of the methodology and presentation of results (some figures also seem to be missing). Reviewer 2 has a deeper concern that, by allowing innovations to occur along only two cultural lineages the models artificially increase the probability that all cultural products emerge from the same lineage. If possible, additional simulations would be very useful to address this concern.

Authors' reply: We are grateful for such a welcoming perspective on the relevance and timeliness of our questions and approach. We appreciated the reviewers' comments as they helped us to significantly improve the clarity and presentation of our findings. We were particularly grateful for the critical point raised by Reviewer 2 on the set up of our models.

We understand that, relative to a case where several cultural lineages are considered, our models accounting for two cultural lineages may increase the chance that all cultural products all emerge from the same lineage. The reviewer suggested two ways around this – to either develop further models to allowing innovations to occur along multiple cultural lineages, or acknowledge the potential limitations of our approach. After carefully consideration, we opted to rephrase our discussion section to give full consideration to the influence of stochastic events in the context of the number of cultural lineages. Our decision was based on (i) the technical implementation of multiple lineages in our current models, and (ii) the original scope of our study.

In revisiting the structure of our agent-based models, it became clear that adjusting the code to include multiple cultural lineages implies a series of new design decisions that are not trivial. For instance, one needs to decide the rules for the agents to integrate inventory items of multiple lineages, and how to design the payoff structure for the resultant high-value products. Such decisions will have an impact on the model outcomes, therefore will require an entirely new model to accommodate them. Our concern is that such changes in design can cause their outcomes to be incomparable with our previous models, which were themselves built upon empirical and theoretical research that considered only 2 cultural lineages. With an entirely new model, and new sets of results and figures, we would require more room for discussion, which is beyond the scope of the current manuscript, and obscure our main message on the effects of optimal levels of connectivity varying with population size and social architecture.

We believe that the point raised by the reviewer can only be appropriately tackled in a follow-up study. Therefore, we opted to follow their second suggestion. We have carefully revised our Discussion section to add a full paragraph (L343-359) that (i) acknowledges the limitation of our approach with few cultural lineages when discussing the role of stochasticity on cumulative culture evolution, but also (ii) provides avenues for future work aiming to develop specific models that verify the relationship between stochasticity and diversity of cultural lineages.

COMMENTS BY REVIEWER 1

Comment#1 by Reviewer 1: From the outset of cultural evolution research understanding the transmission of information within groups and populations has played a major role. In recent years, observational and theoretical work has shown the importance of different aspects of this complex process on cultural dynamics, from transmission modes to population size and connectivity. Only recently have actual social network aspects moved into the field's focus. The present manuscript provides a systematic study of different network types, network sizes, and network connectivity, and their combined and individual effects on the emergence of new traits and their spread in a population.

I enjoyed reading this manuscript. In my opinion, the topic is timely and relevant to the field. There is very little to criticise about the overall presentation of the work both in text and figures. The Background provides relevant information. The methods are (mostly) written clear and concise. The results are generally well presented and discussed.

Authors' reply: Thank you for the very positive feedback highlighting the strong points of our work. We appreciate the attentive review and useful suggestions, each of which we addressed as follows. Note that line numbers refer to the version with in-line tracked changes.

Comment#2 by Reviewer 1: ll. 163-168 "For each population size, we used the Cumulative Incidence Function to estimate the proportion of simulations in which agents reached the recombination of each cultural lineage's products into a final high-payoff product. We used the nonparametric Kaplan-Meier product limit estimator to represent the time intervals based on observed recombination events from 5,000 simulations from model 1, calculating 95% confidence intervals with the Greenwood estimator." – Compared to the otherwise detailed description in the methods, there are three concepts mentioned here that should probably receive a little bit more attention. Why do the authors use them, how do they work, and/or what do they mean?

Authors' reply: We appreciate the opportunity to better justify the choice of our methods. We now explain that the outcomes of our simulations can be interpreted as 'time-to-event' data, therefore are suitable to be analysed with tools from 'survival analyses', such as the Kaplan-Meier estimator. Analogously to data from medical research, in which a certain treatment can be evaluated by the proportion of patients living after a time lag, in our case we were interested in the time taken to reach cultural recombination (the 'event') across population sizes, social network architectures and densities of connections (the 'treatments'). We have rephrased and improved this excerpt (L163-171) as follows:

"To compare time to recombination, we used time-to-event (survival) analyses [25] where time to recombination was a function of network architecture and connectivity. Our simulations yielded time-to-event data, so we used the following methods from survival analysis, which are suitable for such data. For each population size, we used the Cumulative Incidence Function to estimate the proportion of simulations in which agents reached the recombination of each cultural lineage's products into a final high-payoff product (the 'event'). We used the non-parametric Kaplan-Meier product limit estimator to estimate the 'survival function' from this time-to-event data; since we represented the time intervals based on observed recombination events from 5,000 simulations from model 1, we also calculated the 95% confidence intervals (using the Greenwood estimator)."

Comment#3 by Reviewer 1: l. 270 "Fig. S5" – There is neither a Figure 5 nor a Figure S5 in the files I could access

Authors' reply: Excuse our mistake: we opted to replace the Figure S5 with the Table S2 prior the submission and forgot to update the main text. Please note that the coefficient estimates in Table S2 indicate that population size, in particular, causes the most variation in outcomes, compared to network architecture or connectivity. For example, the estimate for population size $N=324$ is 0.618, which means that time to recombination was reduced by about 40% on average – this difference in means far exceeds any other differences. We have fixed this typo, and improved both the Table S2 and the sentence to make our point clearer (L273-275):

“When evaluating performance from the start of the simulations until the time to diffusion, population size caused the most variation in outcomes, compared to network architecture or connectivity (Table S2).”

Comment#4 by Reviewer 1: ll. 322-324 "Our simulations demonstrate that the contribution of large differences in connectivity outweighs any effects pertaining to architecture, at least when information is broadcast (i.e. a one-to-many diffusion mechanism)." – Maybe rephrase. What do the authors mean by the contribution of large differences in connectivity?

Authors' reply: By “large differences in connectivity” we meant the large variance in the social connectivity of our simulated networks: from 8 to 30 social connections per individual. We have rephrased this excerpt to clarify our point (L326-339):

“Fully connected networks have been commonly used to evaluate the performance of a transmission network with a given set of characteristics [3,19]. However, a fully-connected social network representing a human population of any reasonable size would represent an unrealistically high level of connectivity [40], even in the fractal-like human social networks [41]. In addition, our simulations demonstrate that the contribution of large differences in connectivity to the rates of CCE outweighs any effects of network architecture, at least when information is broadcast (i.e. a one-to-many diffusion mechanism). Thus, we suggest that fully connected networks are uninformative null models for testing the influence of social structure on the tempo of CCE. Instead, random networks of similar sizes and densities of connections as the network of interest would provide a more robust benchmark [see also 42]”.

Comment#5 by Reviewer 1: Figure 4 was confusing to me. What are the y-axes showing? In B-D there are too many lines colours and shades, the message is not clear, it is neither intuitive to grasp what the authors want to convey here, nor is it covered by the figure's caption.

Authors' reply: Thank you for this comment that motivated us to rethink the message that Figure 4 conveys. Its take-home message is that stochasticity in early events can affect cultural diversity, and therefore the outcomes of cumulative cultural evolution, even within the same network architecture. We have tracked the diversity of the cultural products (single item, dyad, triad, recombination) across time in a fully connected social network in comparison to a multilevel network. Following from the results in Figure 3, we show in Figure 4 that stochasticity can lead to distinct cultural trajectories among the highly structured networks: note that the distribution of time to reach recombination is typically bimodal in these highly structured networks. The y-axis of the other plots exemplifies some of these trajectories via the proportion of the population that acquired one of those 4 cultural products.

We agree with the reviewer that the original layout of the Figure 4 may have hindered these messages, particularly due to the many overlapping lines, colours and shades. We have now simplified and replotted this figure to distil our message better. We have:

- (i) omitted the distribution of the single inventory item (but still present it in the supplementary figure; see next comment);
- (ii) reduced the types of lines, from 4 to 2 (full and dashed);
- (iii) reduced the number of shades, from 4 to 2 colours (grey and yellow);

- (iv) facet the plots to separate the distributions of each cultural lineage.
- (v) changed from a 2x2 to a 1x4 plot to clarify the labels of the y-axes.

Finally, we have rephrased the figure caption not only to follow these changes but also to facilitate the interpretation of the results (L556-571), as per our explanation above.

Comment#6 by Reviewer 1: Figure S1, the different types of lines are hard to differentiate, maybe consider using different colours for the four possible traits (instead of using it for the different network types, which are already clearly delineated by the graph plots and the headings).

Authors' reply: We appreciate the suggestions to improve the presentation of this figure. We agree with the reviewer that in this supplementary figure we do not need to follow the colour code for the different network architectures used in the main text. We have now prepared a new version of this supplementary figure, which was improved by:

- (i) setting a simpler colour code with high contrast (red vs. black) to highlight the differences between the two cultural lineages (A and B);
- (ii) splitting the plots per cultural lineage so to have fewer lines per plot;
- (iii) increasing the distinction between the line thicknesses and types (thin full; medium dotted, medium dashed, thick full lines) to highlight the differences between the cultural traits (single item, dyad, triad, recombination);
- (iv) improving the readability of the plots by adding a label to the y-axis (the proportion of the population with a given cultural trait) and a legend at the bottom to point out clearly to all types of cultural traits;
- (v) rephrasing the figure caption to be more self-explanatory.

Comment#7 by Reviewer 1: The deposited data, code, and results on GitHub are a great addition to the manuscript. (Note: I noticed that some files are missing, e.g. output for model 1 and model 2 in 3_R_agent_based_models, and the code for figures 1,2, and 4 in 4_create_figures). If possible, and to improve replicability, I would suggest adding even more comments, especially to the simulation code.

Authors' reply: Thank you for checking the repository and spotting these inconsistencies. We have (i) added the missing files to replicate the four main figures and the supplementary figure, (ii) improved the 'ReadMe' files to be more informative, and (iii) annotated the main Python and R codes better by adding several explanatory comments to make the agent-based models more user-friendly. Please note that we intended to leave the "output" folders (now called "results") empty as they serve as storage for the models outputs once the user runs them. However, we appreciate that the model is computationally expensive and not all readers will want to run the code to replicate our results. Thus, we have now added the raw dataset from our own runs (now in the "data" folders) to go along with the code to plot the figures. Please see the updates at https://github.com/simeonqs/Social_network_architecture_and_the_tempo_of_cumulative_cultural_evolution

COMMENTS BY REVIEWER 2

Comment#1 by Reviewer 2: This paper presents two agent-based models that explore how network architecture affects cumulative culture evolution. Six different network architectures (random, small-world, lattice, modular, modular lattice, and multilevel) capturing different levels and combinations of clustering and modularity have been implemented in populations with different sizes and densities of connections. The simulations confirm recent results that combinatorial innovation is optimized at intermediate densities of connections but also reveals that the optimal level of connectivity varies with population size. Although not entirely surprising, this result is important to the current debate regarding whether humans' unique multilevel social structure accelerates cumulative cultural evolution. The comparison between the two models implementing either one-to-many or one-to-one diffusion processes also reveals interesting results and highlight the complex interaction between population size, structure and transmission mechanisms.

My opinion is that, although the paper does not yield any major finding, it provides a lot of food for thought to the large community of scholars interested in the relationship between demography and cumulative culture. The goal of the paper of disentangling the relative contribution of network architecture from those of connectivity and population size is clearly an important one.

Authors' reply: We very much appreciate the accurate summary of our work, as well as the reviewer's opinion on its novelty and relevance.

Comment#2 by Reviewer 2: My only concern is that the authors limited their investigations to a situation where innovations of cultural products can only take place along two cultural lineages. I don't think this threatens the validity of their main result (that is optimal level of connectivity varies with population size) but this clearly increases the chance that cultural products all emerge from the same lineage. I would expect the results of their simulations to be less influenced by stochastic events were the fitness landscapes to include more cultural lineages. I would recommend the authors to either run additional simulations to verify this relationship or explicitly mention this limitation when they discuss the role of stochastic events on cumulative culture.

Authors' reply: We are very grateful for this comment, which we considered carefully. We agree that this is an important point to address, but after careful evaluation, we decided that encoding additional cultural lineages in the current models would require making too many decisions that would distract from central aims of the present manuscript. These include how to change the payoff structures to integrate additional lineages, which products can be recombined, etc. The appropriate implementation of multiple lineages would require an entirely new model, which would need a substantial amount of space to develop and fully explore. Fully exploring this model would draw attention away from providing a clear message, which both reviewers got very clearly from the present manuscript. Since we agree with the Reviewer that considering two lineages – as done by the original studies that inspired our models: Derex & Boyd 2016, Migliano et al. 2020 – does not invalidate our main results, we have opted to follow the Reviewer's second suggestion. We have amended the Discussion section to acknowledge the limitation of our current approach with the few cultural lineages, but also to emphasise that future work should aim to develop specific models that verify the relationship between stochasticity and diversity of cultural lineages and how this relationship can impact cumulative culture evolution. These changes are now highlighted in a full new paragraph (L343-359):

“One major remaining question is how the diversity of cultural lineages might affect CCE across different network architectures. Our simulations were built on previous empirical and theoretical work that considered two cultural lineages [12,19]. Increasing the number of lineages could potentially reduce or increase the time to recombination, depending on how the recombinations have to be made and on their

pay-offs. For example, recombinations from any two of three or more lineages would likely emerge faster, as the probability that any two lineages survive is higher. By contrast, recombinations requiring products from three lineages could take substantially longer, especially in more connected networks (or with one-to-many diffusion dynamics), since all three would have to become established. However, adding more lineages would be unlikely to reduce the influence of stochasticity in any given diffusion. This is especially the case in highly connected networks or one-to-many diffusion dynamics, because it would not stop one lineage from initially dominating (when a single invention spreads throughout the population, before another lineage is invented, see Fig. 4). Rather, the tendency for one lineage to dominate would be more heavily impacted by the payoff structure associated with incremental improvements to products. While we do not believe that increasing the number of lineages would substantially change the primary findings of this study, a promising avenue for further research would be a more in-depth exploration of how exposure to multiple cultural lineages may shape the tempo and mode of CCE.”